# Comprehensive Time-Course Effects of Combined Training on Hypertensive Older Adults: A Randomized Control Trial

**DOI:** 10.3390/ijerph191711042

**Published:** 2022-09-04

**Authors:** Amanda V. Sardeli, Arthur F. Gáspari, Wellington M. dos Santos, Amanda A. de Araujo, Kátia de Angelis, Lilian O. Mariano, Cláudia R. Cavaglieri, Bo Fernhall, Mara Patrícia T. Chacon-Mikahil

**Affiliations:** 1Laboratory of Physiology of Exercise, Scholl of Physical Education, State University of Campinas, Campinas 13083-851, SP, Brazil; 2Gerontology Program, Scholl of Medical Sciences, State University of Campinas, Campinas 13083-888, SP, Brazil; 3Institute of Inflammation and Ageing, University of Birmingham, Birmingham B15 2WB, UK; 4Sidia Institute of Science and Technology, Manaus 69055-035, AM, Brazil; 5Physiology Department, Federal University of Sao Paulo UNIFESP, São Paulo 04023-901, SP, Brazil; 6Laboratory of Translational Physiology, Nove de Julho University, São Paulo 01525-000, SP, Brazil; 7Integrative Physiology Laboratory, University of Illinois at Chicago, Chicago, IL 60608, USA

**Keywords:** aging, hypertension, exercise, blood pressure, cardiorespiratory fitness, resistance training

## Abstract

The aim was to identify whether 16 weeks of combined training (Training) reduces blood pressure of hypertensive older adults and what the key fitness, hemodynamic, autonomic, inflammatory, oxidative, glucose and/or lipid mediators of this intervention would be. Fifty-two individuals were randomized to either 16 weeks of Training or control group who remained physically inactive (Control). Training included walking/running at 63% of $\dot{V}$O_2_max, three times per week, and strength training, consisting of one set of fifteen repetitions (seven exercises) at moderate intensity, twice per week. Both groups underwent a comprehensive health assessment at baseline (W0) and every four weeks, for 16 weeks total. *p*-value ≤ 0.05 was set as significant. Training did not reduce blood pressure. It increased $\dot{V}$O_2_max after eight weeks and again after 16 weeks (~18%), differently from the Control group. At 16 weeks, Training increased strength (~8%), slightly reduced body mass (~1%), and reduced the number of individuals with metabolic syndrome (~7%). No other changes were observed (heart rate, carotid compliance, body composition, glycemic and lipid profile, inflammatory markers and oxidative profile, vasoactive substances, heart rate variability indices). Although Training increased cardiorespiratory fitness and strength, Training was able to reduce neither blood pressure nor a wide range of mediators in hypertensive older adults, suggesting other exercise interventions might be necessary to improve overall health in this population. The novelty of this study was the time-course characterization of Training effects, surprisingly demonstrating stability among a comprehensive number of health outcomes in hypertensive older adults, including blood pressure.

## 1. Introduction

Aging increases the chance of become hypertensive and, in Brazil for example, 64.2% of individuals above 60 years old were hypertensive in the last assessment in 2016 [1]. Despite, some reports of hypotension, syncope, and medication overload in high-risk patients when the treatment target for systolic blood pressure (BP) is under 120 mmHg; the maintenance of systolic BP ≤ 120 mmHg [2,3].

Although the antihypertensive effects of different types of exercise training in the general population are well known [4], the ability of exercise to reduce BP in older adults is still unclear [5]. Understanding how exercise affects this heterogeneous population is fundamental, not just as most older adults are taking medication that impair the isolation of exercise effects on the controlled BP [6,7], but also as treatment of hypertension in this population is expected to be more complex (hypertension in older adults could be a combination of different aging-related changes), and training-mediated BP alterations differ between younger and older adults [8].

Aerobic training is recommended for treatment of hypertension, but emerging research suggests dynamic strength training can also be beneficial [5]. A recent meta-analysis showed just one study which tested the combination of aerobic training and strength training (i.e., combined training) effects on hypertensive women 65 years or older [9]. In the same meta-analysis, within the larger sample of hypertensive older adults ≥ 50 yrs, combined training was not able to reduce systolic BP [9] (which agreed with another meta-analysis in the general population [4]). Furthermore, the overall exercise effects that included other types of training were more dependent on baseline BP levels than age.

The mechanism mediating the BP reduction with exercise training is still unclear. However, inflammation and autonomic modulation are suggested to be mediators of exercise training effects on BP [10]. In spontaneously hypertensive rats, two weeks of aerobic training reduced inflammation in the hypothalamic nuclei of cardiovascular control and increased baroreflex sensitivity [11]. After four weeks, parasympathetic modulation of the heart increased and sympathetic modulation of the arteries decreased; finally, after eight weeks, BP was significantly reduced [11]. Although animal results shed lights on the possible mechanism mediating the BP reduction following exercise training, they are not representative of hypertensive older adults developing hypertension along their life span [12]. Thus, the time-course of all these adaptations triggered by combined training remains unknown in humans.

The present study aims to investigate the effect of 16 weeks of combined training (Training) primarily on BP in hypertensive older adults who were maintained on their medication. Second, we investigated potential key fitness, hemodynamic, autonomic, inflammatory, oxidative, glucose and/or lipid mediators of these adaptations across time (month by month). We hypothesized that reduction in inflammatory markers would be followed by autonomic modulation improvements, which in turn would precede a blood pressure reduction in this population similar to that observed in rats [11]. However, since hypertensive older adults, unlike spontaneously hypertensive rats, develop hypertension in association with many other health issues, we did not discard the possible impact of other factors [10].

## 2. Materials and Methods

### 2.1. Experimental Approach to the Problem

This study was part of a prospective interventional randomized controlled trial, with a parallel control group, executed in Campinas (São Paulo, Brazil) registered on the Brazilian registry of clinical trials ([https://ensaiosclinicos.gov.br/rg/RBR-3yxds4], identifier [U1111-1181-4455]). The sample size calculation (G*Power 3.2.1 software), based on mean blood pressure changes with Training, ensured the number of participants (n = 46) analyzed at the end were above that necessary for 95% power, as estimated previously [10].

### 2.2. Participants

The 52 subjects were listed from high to lower values of their pre intervention assessments for the five primary variables (age, sex, systolic BP (SBP), body mass index (BMI) and RR interval (RRi)); then they were pair matched to be drawn to group 1 (Training) or 2 (Control), using a computerized random function with participants blinded and aware that they could be assigned to any group, which would receive Training during the study or after the study (Control). To ensure proper balance, we continuously inverted a few participants’ matched pairs until the baseline difference between groups reached an effect size smaller than 0.2 (Cohen’s d) for each of the five primary variables.

Eligible participants were men and women over 60 years old recruited from the community. Table 1 describes the baseline characteristics of the 46 participants that completed the study, and the balance achieved at baseline assessments between groups was not maintained (ES < 0.2) for BMI and RRi. None of the subjects were physically active (<150 min of exercise/week), however we did report their specific Physical activity level (PAL) at baseline [10]. All subjects were diagnosed with hypertension by their physician and were currently taking antihypertensive medication (Table 1). Beta-blockers were considered an exclusion criterion due to the direct influence on heart rate, and due to participants having acute anti-inflammatory medication administered, their tests were delayed until one week after medication cessation. Participants were invited by radio, television, websites, delivering flyers around the University area and by phone calls to the laboratory database. After initial screening, pre-approved participants underwent a clinical evaluation by a physician, who cleared them for participating in physical activity (general physical examination, cardiological and clinical exercise testing). Exclusion criteria were BMI > 35 kg/m^2^, resting SBP > 170 mmHg or diastolic BP (DBP) > 110 mmHg, coronary artery disease, insulin-dependent diabetes mellitus, chronic obstructive pulmonary disease, osteoarticular disease that could limit participation in the exercise training program, and cigarette smoking. All selected individuals signed the informed consent approved by Ethics Committee from the University.

### 2.3. Intervention

The 16-week Training protocol was based on the American College of Sports Medicine recommendations for older adults [13] using an aerobic training intensity proposed for hypertension treatment [5]. Considering aerobic training is the most common exercise for cardiovascular improvements, including blood pressure reduction in hypertensive older adults [5,9,14], a greater portion of the training volume was composed of aerobic training rather than strength training. All exercise sessions were individually prescribed and supervised. Twice a week, on Mondays and Thursdays, strength training (15 min duration) was followed by the aerobic training (50 min duration) and once per week, on Fridays, aerobic training was performed alone (50 min duration), with the protocol consisting of continuous walking and/or running on a treadmill. Strength training consisted of one set of 15 repetitions for each of seven strength exercises for the major muscle groups (leg extension and flexion, leg press, heel lift, bench press, pulley, and abdominal). To avoid the exposure of the participants to repetition maximum tests [15], the strength training load was adjusted according to an individual’s perception of every training session to achieve a moderate intensity using the Borg scale (5–6 on a 10-point scale) [13]. The participants were familiarized with the 10-point scale in the first week of training. Aerobic training was prescribed at 63% of the $\dot{V}$O_2_max recommended for hypertensive individuals, and adjusted after 8 weeks, based on the new test. Participants in the control group (Control) did not receive any treatment. All participants, including the Control, were advised to maintain their normal diet and all their prescribed medications during the 16 weeks of intervention. In addition, the monthly assessments facilitated the engagement of the Control on the research, since they were frequently followed by the researchers.

### 2.4. Outcomes

The main outcomes were assessed in both groups at baseline, 4, 8, 12 and 16 weeks of control or training intervention, for primary (BP) and secondary variables (all other variables) as described in the protocol registration [10] and in the methods section. Each participant was tested at the same time of day and with the same evaluator at each of the 5 testing time points. At least 48 h rest between the prior training session and the assessments was maintained, and the participants were asked to fast for 12 h before the cardiovascular function assessments and the blood draw. The participants were monitored for current infections and medication usage every month by questionnaires [10]. When there was risk of infection or the participants forgot to take their standard medication, the monthly assessments were rescheduled for the next week. Although participants, training instructors, and evaluators were not blinded during data collection, the overall data exported for further computational (e.g., ultrasound images or BP time interval data) and statistical analysis were blinded.

*Physical fitness (Maximum oxygen consumption (*$\dot{V}$O_2_max*), maximum strength and functionality**)**:* The maximal exercise test was performed on a treadmill, with breath-by-breath gas analysis (CPX, Medical Graphics, Saint Paul, MN, USA). The protocol began with a 1% incline, starting at 4 km/h (2 min), followed by increases of 0.3 km/h every 30 s until volitional exhaustion. Attainment of two of the following criteria were used to ensure only tests with peak effort were used in the analysis: (1) Respiratory exchange ratio > 1.15, (2) at least 10 bpm below the maximum heart rate predicted by age (220 minus age), (3) plateau in the oxygen consumption (increase < 100 mL/min) even with an increase in intensity. With or without plateau, the highest 30 s average was registered for analysis. A re-test after 48 h was performed to avoid bias.

Peak torque for isometric strength at 60° of flexion (0° = complete extension) of leg extension exercise and isokinetic strength (60°/s) of concentric contractions of leg extension and flexion exercise, were assessed using the Biodex System 4 isokinetic dynamometer (Biomedical Systems, Newark, CA, USA). Participants were seated with their preferred leg knee rotation center (middle intercondylar line) aligned with the center of rotation of the device arm, which was set at an angle of 60° from the horizontal plane. The support point of the lever arm of the device was located one centimeter proximal to the medial malleolus of the participant, and trunk and hips were tied with straps to avoid auxiliary movement. The warm-up was composed of 10 submaximal isometric leg extension and leg flexion contractions, with 3 s duration and 10 s intervals between them. After warm-up, participants performed 3 sets of isometric leg extensions and flexion with 30 s intervals between the movements (extension and flexion) and 2 min between sets.

Data collected in the third attempt of isometric leg flexion and extensions were used for force development rate analysis. The time taken from the beginning of the test to the point of maximum contraction was entered into the equation (TDF = Δ Strength/Δ time). The test start time was considered when there was a variation greater than 7.5 N from baseline.

The isokinetic warm-up was composed of 5–6 passive repetitions and 3 submaximal contractions. Participants performed 5 isokinetic contractions with maximum knee extension/flexion contractions, at a speed of 60°/s. In all assessments, volunteers were instructed to produce the maximum amount of force, as quickly as possible, and received verbal encouragement. The highest torque value (torque peak) in each test was recorded. Handgrip strength was quantified through the best of three trials, with the dominant hand using a Jamar dynamometer (Lafayette Instruments, Lafayette, IN, USA). For all tests, participants were asked to use maximal effort, as fast as possible, and they received verbal encouragement [10]. The best of three trials was registered.

Functionality was assessed with a stand and sit test (30 s chair standing), timed up and go, sit and reach test, gait speed along 4.6 m and Balance, according to classical assessments described in the research protocol [10].

*Body composition:* The weight will be measured by a calibrated scale (digital scale Filizola^®^, São Paulo, Brazil, model ID1500) to the nearest 100 g. Height will be measured to the nearest 0.5 cm using a stadiometer (Digital Filizola^®^). BMI will be calculated from these values (weight/height^2^). The waist, hips and neck circumference will be measured by tape, as reliable markers of cardiometabolic risk. Body volume was assessed through a densitometric technique at a plethysmograph chamber. The evaluation instrument is the chamber plethysmography (BOD POD^®^) connected to a software that determines the air volume variation and interior pressure from when it was empty to when the participant was there, and variables necessary for estimating body volume. From the body volume the lean and fat mass were estimated using Siri’s equation [16]. A 10 to 5 MHz linear transducer coupled with ultrasound (Nanomaxxtm, SonoSite, Bothell, WA, USA) were used for assessment of rectus femoris and vastus lateralis thickness [10]. The distance from the femoral trochanter and the lateral condyle of the tibia serves as a guide for latero-lateral location, while the distance from the distal second third of the trochanter serves as a basis for locating the anterosuperior point, composing the desired point for image capture of the vastus lateralis. This point is transferred to the anterior part of the thigh and a new measurement is taken from the distance from this point to the base of the patella where the image of the rectus femoris was captured. The acquired images were analyzed on the ImageJ software, and the pixels were converted into centimeters.

Cardiovascular assessments (Hemodynamics, Ankle brachial index, Heart rate variability, Carotid compliance, Intima Media Thickness): Following 15 min supine rest, the BP was assessed in supine position using an aneroid sphygmomanometer in the right arm, three times (5 min interval between each assessment) and the mean was used for analysis, and the assessment of systolic upper and lower limbs BP were used for ankle brachial index (ABI) calculation following standardized recommendations, as previously described [10]. Beat-to-beat SBP and DBP were obtained using finger photoplethysmography by Finometer Pro^®^. The average of 300 beats at a stationary period in supine rest was used to estimate stroke volume, heart rate, total peripheral resistance, cardiac output, and baroreflex sensitivity.

Furthermore, in the supine position, the resting range of the respiratory rate was confirmed (9–22 breaths/min) and continuous R-R intervals were acquired by a heart rate monitor. Heart rate variability (HRV) was analyzed in both time and frequency domains, for five minutes stationary R-R intervals, in Kubios HRV analysis software.

Using a linear transducer of 10–5 MHz and the ultrasound M mode, a sequence of images of the left carotid common artery diameter (2 cm proximal to the carotid bifurcation) was acquired for 5 s. These images were used for Carotid compliance (CC) calculation and the intima media thickness of the far wall was evaluated as the distance between the lumen-intima interface and the media-adventitia interface [10].

*Blood markers:* Samples of serum, heparin- and EDTA-plasma were obtained after 12 h fasting, drawn from an antecubital vein and stored at −80 °C. Glucose tests were conducted immediately. Glucose and lipid analyses were performed by standard method in Clinical Laboratory, in which triglycerides (TG) and total cholesterol (TC) were obtained by enzyme-trinder method, HDL by selective detergent method and LDL by using the Friedewald equation (LDL = (TC − HDL) − (TG/5)). Insulin, C-reactive protein (CRP), adiponectin and leptin, and the vasoconstrictor endothelin-1 (ET-1) were assessed by multiple analytics magnetic assay, while ultrasensitive Interleukin 6 (IL-6), tumor necrosis factor alpha (TNF-α) and the vasodilator nitrite were assessed using ELISAs (enzyme-linked immunosorbent assay) with a plate reader, and both methods using kits (R&D Systems). Insulin resistance was calculated according to the equation HOMA-IR = fasting insulin (µU/mL) fasting glucose (mmol/L)/22.5 [17]. We quantified hydrogen peroxide, due to its high oxidative potential, as well as carbonyls, as markers of protein peroxidation, and thiobarbituric acid reactive substances (TBARS) as a marker of lipid peroxidation. For this, the total proteins of the sample were determined using bovine serum albumin as a standard. Approximately 8 mg/mL of protein was analyzed for oxidative stress assessments. Furthermore, to assess the antioxidant potential of the sample, we quantified the activity of NADPH oxidase and superoxide dismutase (SOD) enzymes, and the ferric reduction ability power (FRAP), as a marker of global antioxidant capacity. These analyses followed methods previously described [18]. In summary, the method to quantify carbonyls uses the reaction of carbonyl groups with 2,4-dinitrophenylhydrazine (DNPH) to form, 2,4-dinitrophenylhydrazine, and was measured spectrophotometrically at 360 nm.

For the TBRAS reaction we added 150 μL of Dodecyl Sulfate of Sodium (SDS) at 8.1% (*w*/*v*), 300 μL of Trichloroacetic Acid (TCA) (Vetec Quimica Fine Ltd., Xerem Duque De Caxias, RJ, Brazil) at 20% (*w*/*v*) and 500 μL of Thiobarbituric Acid (Sigma-Aldrich Corporation, Gillingham, UK). This mixture was incubated for 20–30 min at 95 °C, forming a pinkish compound, and then was cooled on ice. After this procedure the tubes were centrifuged at a speed of 4000 rpm for 5 min (Eppendorf AG, Germany) and 200 μL of supernatant was added to an Elisa plate well. The reading was taken at 535 nm using an Elisa Plate reader. Hydrogen peroxide was measured by red oxidation of phenol, catalyzed by radish peroxidase (PRS), at 630 nm. A volume of 70 μL of plasma together with 180 μL PRS were incubated for 25 min at room temperature. After this period, 5 μL of NaOH was added and the reading was performed using an Elisa’s plate reader. The quantification of hydrogen peroxide was estimated based on a standard curve. The activity of the NADPH oxidase enzyme was determined in blood plasma and was evaluated by the production of superoxide determined by ELISA. For performance of the assay, a 50 mM phosphate buffer containing 2 mM EDTA and sucrose 150 mM, 1.3 mM NADPH and 10 μL sample was used. Regarding FRAP, insofar as any sock reaction having a lower redox potential under reaction conditions than a half ferric/ferrous reaction, this would convert the ferric (FeIII) reaction to a ferrous (FeII) reaction. Therefore, the change in absorbance is directly related to the power of total donation reduction in antioxidant electrons present in the reaction. After incubating 10 μL of the sample and 290 μL of the FRAP reagent (Acetate Buffer sodium and acetic acid pH 3.6; 10 mM Tripiridil 2,4,6-S-Thiazine Solution; Solution of ferric chloride hexahydrate) for 5 min with a shake at 37 °C. The reading was performed at 593 nm. The analysis of SOD is based on the inhibition of the reaction of the superoxide radical with pyrogallol. A unit of SOD is defined as the quantity of enzyme that is inhibited by 50% of the oxidation rate of the detector. The oxidation of pyrogallol leads to the formation of a colored product, detected spectrophotometrically at 420 nm for 2 min, and the SOD activity was determined by measuring the velocity of formation of oxidized pyrogallol.

*Metabolic syndrome:* According to the United States national cholesterol education program [19], participants were considered positive for metabolic syndrome (MetS) if they were positive for three or more of these five criteria: High waist circumference: women > 88 cm and men > 102 cm; Hypertension: SBP > 130 mmHg, or DBP > 85 mmHg, or using anti-hypertensive medication; Fasting blood glucose > 110 mg/dL or anti-hyperglycemic medication; High triglycerides: >150 mg/dL; low HDL: women < 50 mg/dL and men < 40 mg/dL.

### 2.5. Statistical Analysis

Analyses were performed using SPSS version 24. Between-group baseline characteristics were compared by independent *t*-test. First, the Shapiro–Wilk test was used to assess the normality of the distribution for each variable. The non-normally distributed data were transformed by logarithm ([Log]), or square root ([SqR]) transformations, or the whole Box–Cox family ([Box–Cox]). All data were described in the text and tables in raw format as mean ± standard deviation; additionally, we tagged the variables as [Log], [SqR] or no tag (when raw data was already normally distributed), according to their required transformation to become normally distributed for analysis. When none of the transformations led to normal distribution, we analyzed the raw format and target it as [NN] (non-normally distributed).

Mixed-model analyses were conducted for all variables with groups (Training and Control) and time points (W0, W4, W8, W12 and W16) as fixed factors and participants as random factors (intercept and curves). The first order autoregressive covariance model was used, considering a progressive change was expected along in participants with time. When there was significant group*time interaction (*p* < 0.1), Bonferroni post-hoc was applied and *p* < 0.1 was also accepted as significant.

Following the identification of outliers (score z > 2.58 or <−2.58 and out of physiological variability) the analyses were performed with and without outliers. The results based on analyses with outliers removed ([OR]) were presented only when there were significant differences between or within groups that were not found through analyses with outliers (TBARS and CC).

## 3. Results

Twenty-three individuals were analyzed in each group (Figure 1), representing 88.46% of the randomized subjects. Participants’ eating habits were not altered in each group [20]. There were a few alterations in participants’ medication used: while physicians of four Training participants required them to stop their medication (one diuretic, one herbal medicine and two antidepressant), two Control participants had to increase their doses of metformin and statin. No adverse events were reported during the study and the exercise training prescriptions were well tolerated. When the participants would miss a session, they were asked to participate in an extra session before the next monthly assessments, thus all included participants completed at least 10 training sessions per month.

Training exhibited reduced body mass and BMI at W16 compared to W0, W4 and W8, but this lower value (at W16) was not significantly different from Control. Control exhibited an increase in waist circumference at W12 compared to W0, but this was not different from Training (Table 2). No interaction effect was observed for our primary outcome (i.e., BP), and only significant time effects for SBP and ABI, with a reduction in SBP and an increase in ABI from W0 to W12 were observed (Table 2). There was a significantly lower number of individuals with Metabolic Syndrome (MetS) in Training compared to Control only at W16, according to Chi squared test (Table 2).

The Training group showed higher isometric and isokinetic knee extension than Control at W16 (Table 3). While there was no significant increase within Training, there was a reduction in isometric and isokinetic knee extension, and in the rate of force development within Control at W16. $\dot{V}$O_2_max and maximal speed increased within Training from W0 to W8 and again from W8 to W16, which was significantly different from Control at W16. No group*time interaction was observed for the functionality tests, but significant time effects were observed from W0 to W16; showing improvements for stand and sit, time up and go, and gait speed for both groups.

No change occurred for any of the blood markers such as glycemic, lipidic, inflammatory, oxidative stress markers or vasoactive substances over the course of the intervention (Table 4).

There were no significant effects for any of the time domain, frequency domain or even the non-linear HRV indexes (Table 5).

## 4. Discussion

There is a strong body of evidence favoring aerobic training for blood pressure reduction in hypertensive individuals compared to combined aerobic and strength training [9]. However, combined training is fundamental to improve comprehensive health needs of older adults with hypertension [21]. Here, Training led to important benefits such as the considerable increase in cardiorespiratory fitness (~18% $\dot{V}$O_2_max), reduction in the number of individuals with MetS compared to Control (24% lower) and modest improvements in strength (~8%) and body mass (~1%).

We attribute the considerable increase in cardiorespiratory fitness primarily to the higher volume of aerobic training proportional to strength training within the Training program (nearly 85% of the time); but also, to the supervision of the training sessions ensuring adherence to the prescribed training intensity. Although, there was a continuous increase in strength load in all exercises throughout the 16 weeks of exercise program [22], the maximum strength tests detected only small strength improvements. The small increase might be due to the low volume and frequency of strength training used in this study compared to others [23,24]. It is doubtful that the exercise prescription method per se, using the rate of perceived exertion (RPE [0 to 10]), influenced our findings considering other studies have shown improvements with RPE prescription. However, since those studies also prescribed strength training with higher volumes and frequencies [25,26], it is not possible to directly compare those results to our findings. Importantly, a meta-analysis of our group [27] showed combined training produces lower increments in muscle strength in hypertensive older adults compared to strength training alone (standardized mean increase of 0.46 [95%CI 0.21; 0.71] and 1.69 [95%CI:1.30; 2.08] for combined training and strength training, respectively). Thus, it is possible that aerobic exercise training, as applied in our study, may partially interfere with strength gains which have also been shown for healthy young adults [28].

Previous studies have shown improvements in the other functional tests such as balance, gait speed, gait quality, and stand and sit with exercise training, including hypertensive older adults; however, the individuals tested in these studies had lower baseline performance than the individuals of the present study [29,30]. It is possible that our subjects exhibited a ceiling effect due to their relatively high functional status at baseline. Thus, the improvements in muscle strength, although of lower magnitude, may have contributed to the functional gains to increase cardiorespiratory fitness.

It is possible the unexpected lack of significance for any of the blood markers and cardiometabolic effects may be explained by the Wilder’s principle, that, applied to our study, would explain the lower effect of Training in individuals with well-controlled clinical markers. Wilder’s principle states that the direction of response of a body function to any agent depends largely on the initial level of that function. Wilder’s principle has also been proposed to explain the notable influence of baseline BP values in other anti-hypertensive therapies and has been confirmed in an older hypertensive population [9]. BP was very well controlled in the population in the present study, compared to studies finding a reduction in SBP and DBP, and this behavior was observed not only for BP but for many other health markers [31]. For instance, the individuals tested in this study had healthier values of TNFα, IL-6, CRP, IL-1ra, adiponectin and leptin compared to individuals from a similar population and the studies showing significant improvements with exercise training were found in individuals with altered baseline values of these markers [32,33,34,35]. Furthermore, it is possible the high level of education (Training:10.18 ± 4.6 and Control:11.17 ± 4.28 years of study) and socioeconomic status of our sample (individuals aging in a well-developed location around the University) contributed to good control of their comorbidities.

There were a few changes in patients’ medication over the course of the study. Since there were only reductions in doses of medication for the Training group, and increase was observed for Control, the contribution of medications to lower BP would be expected to be more persistent in Control and may have contributed to in the lack of an observed training effect. However, considering this population exhibited complex medication regimens and a high likelihood of fragmented care, the reduction in the medication burden observed in Training would be valuable.

It was not possible to confirm our initial hypothesis that immune system and autonomic modulation influences BP reduction, as none of these variables were altered by Training. Thus, this is a limitation of the study and it is not clear if this population may need a different training prescription or other associated interventions to achieve benefits [25]. Regarding the applicability of the trial, the heterogeneity of participants’ responses reinforces the need for individualized prescription that goes beyond the intensity of exercise, as conducted here, but should also be extended to the training components such as variations in volume, frequency and duration of intervention. New clinical trials are already ongoing to test the effectiveness of combined training to reduce BP and other health-related outcomes in hypertensive older adults. One of them is going to test a protocol that is similar to the one tested here [36] and the other one will manipulate different weekly frequencies of combined training for 12 weeks [37].

In addition, it is noteworthy that our sample size calculation was estimated for comparison between two times and two groups, while we analyzed five time points leading to considerably lower power in our analysis. On the other hand, there is no previous controlled trial reporting such detailed time-course of exercise training effects on hypertensive older adults, and this information might add important value to the exercise physiology literature. Specifically, these data highlight the natural effect of time (independent of exercise) that might be considered in future studies.

## 5. Conclusions

Although Training led to important benefits in cardiorespiratory fitness, strength and body mass, specific cardiometabolic variables and blood markers were not improved in hypertensive older adults. Since the within-subject data for most variables were stable throughout in both Training and Control (individual data can be assessed in the Appendix A), and there was high variability between subjects, we suggest studies assessing such adaptations at only one time point, using simpler statistical models, can report random changes caused by type 1 error. Thus, future studies should investigate the effects of different exercise protocols in hypertensive older adults, with robust statistical methods.

## Figures and Tables

**Figure 1 ijerph-19-11042-f001:**
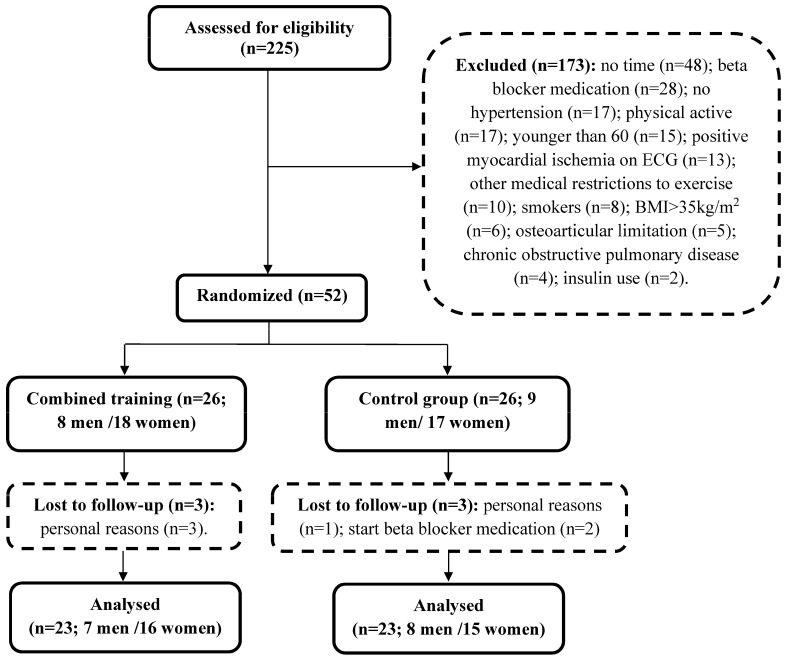
Assessment of eligibility, randomization, and follow-up.

**Table 1 ijerph-19-11042-t001:** Baseline characteristics.

Variables	Training (n = 23)	Control (n = 23)	ES [95%CI]
Age (years)	65.4 ± 4.3	65.2 ± 4.6	0.04 [−8.68; 8.77]
Sex (% of men)	30.43%	34.78	-
Body mass (kg)	78.8 ± 11.4	78.3 ± 13.3	0.04 [−24.17; 24.25]
BMI (kg/m^2^)	29.2 ± 4.0	30.12 ± 3.5	−0.25 [−7.6; 7.1]
MMSE	25 ± 3.3	26 ± 2.9	−0.32 [−6.4; 5.75]
Education (years of study)	10 ± 5	11 ± 4	−0.22 [−9.04; 8.6]
SBP (mmHg)	133 ± 15	132 ± 23	0.05 [−37.19; 37.29]
DBP (mmHg)	86 ± 8.2	79.5 ± 9.5	0.73 [−16.61; 18.08]
RRi (ms)	995 ± 139	916 ± 114	0.62 [−247.32; 248.56]
PAL	6.2 ± 1.6	6.3 ± 1.3	−0.07 [−2.91; 2.77]
**Antihypertensive medication**			
Angiotensin II receptor blockers (n)	5	3	
Angiotensin-converting enzyme inhibitors (n)	12	19	
Calcium channel blockers (n)	3	3	
Diuretics (n)	11	7	
Other vasodilators (n)	1	0	
**Treatments with other drugs**			
Hypercholesterolemia/statins (n)	10	10	
Hyperglycemia (n)	3	5	
Anxiety (n)	2	0	
Depression (n)	4	4	
Hypothyroidism (n)	5	4	
Osteoporosis (n)	1	4	
Osteoarthritis (n)	3	2	
Herbal medicines (n)	6	2	
Vitamin’s complex (n)	4	6	

**Legend:** BMI: Body mass index; MMSE: Mini mental state examination; SBP: Systolic blood pressure; DBP: Diastolic blood pressure; PAL: Physical activity level; Independent *t*-tests showed no baseline difference (*p* > 0.05); ES: Effect Size (Cohen’s d); 95%CI: ninety five percent confidence interval.

**Table 2 ijerph-19-11042-t002:** Cardiometabolic assessments.

Body Composition					
**Body Mass (kg) *p* < 0.001**	**W0**	**W4**	**W8**	**W12**	**W16**
Training	78.9 ± 11.5	78.9 ± 11.8	78.6 ± 11.8	78.2 ± 12.1	77.8 ± 12.2 ^abc^
Control	78.3 ± 13.4	78.5 ± 13.0	78.7 ± 12.9	78.7 ± 12.8	78.7 ± 12.8
ES [95%CI]	0.05 [−24.3; 24.4]	0.03 [−24.2; 24.3]	0 [−24.2; 24.2]	−0.04 [−24.4; 24.4]	−0.07 [−24.6; 24.5]
**BMI (kg/m^2^) *p* < 0.001**					
Training	29.2 ± 4.0	29.3 ± 4.1	29.2 ± 4.2	28.9 ± 4.3	28.8± 4.2 ^abc^
Control	30.12 ± 3.5	30.2 ± 3.6	30.3 ± 3.5	30.3 ± 3.5	30.23± 3.4
ES [95%CI]	−0.24 [−7.6; 7.2]	−0.25 [−7.8; 7.3]	−0.29 [−7.8; 7.2]	−0.36 [−8.0; 7.3]	−0.39 [−7.9; 7.1]
**Fat mass (%) [Box-Cox] *p* = 0.42**					
Training	40 ± 8	40 ± 8	40 ± 9	39 ± 10	39 ± 8
Control	40 ± 9	40 ± 9	40 ± 9	41 ± 9	41 ± 9
ES [95%CI]	0 [−16.7; 16.7]	0 [−16.7; 16.7]	0 [−17.6; 17.6]	−0.21 [−18.8; 18.4]	−0.24 [−16.9; 16.4]
**Fat free mass (%) [Box-Cox] *p* = 0.45**					
Training	60 ± 8	60 ± 8	60 ± 9	61 ± 10	61 ± 8
Control	60 ± 9	60 ± 9	60 ± 9	59 ± 9	59 ± 9
ES [95%CI]	0 [−16.6; 16.6]	0 [−16.7; 16.7]	0 [−17.6; 17.6]	0.21 [−18.4; 18.4]	0.24 [−16.4; 16.9]
**VL thickness (cm) *p* = 0.55**					
Training	3.47 ± 0.72	3.52 ± 0.64	3.54 ± 0.61	3.55 ± 0.62	3.53 ± 0.61
Control	3.39 ± 0.67	3.37 ± 0.59	3.33 ± 0.6	3.33 ± 0.63	3.25 ± 0.62
ES [95%CI]	0.12 [−1.3; 1.5]	0.24 [−1.0; 1.5]	0.35 [−0.8; 1.5]	0.35 [−0.9; 1.6]	0.46 [−0.8; 1.7]
**RF thickness (cm) *p* = 0.53**					
Training	1.38 ± 0.39	1.48 ± 0.37	1.46 ± 0.37	1.45 ± 0.34	1.51 ± 0.4
Control	1.33 ± 0.44	1.34 ± 0.41	1.34 ± 0.42	1.35 ± 0.4	1.32 ± 0.43
ES [95%CI]	0.12 [−0.7; 0.9]	0.35 [−0.4; 1.1]	0.3 [−0.5; 1.1]	0.27 [−0.5; 1.0]	0.46 [−0.4; 1.3]
**Neck circumference (cm) *p* = 0.81**					
Training	38.17 ± 3.25	37.84 ± 3.04	37.92 ± 3.14	37.6 ± 3.02	37.49 ± 3.05
Control	38.44 ± 4.04	38 ± 4.08	38.31 ± 3.8	38.37 ± 3.84	38.36 ± 3.86
ES [95%CI]	−0.07 [−7.2; 7.1]	−0.04 [−7.0; 6.9]	−0.11 [−6.9; 6.7]	−0.22 [−7.0; 6.5]	−0.25 [−7.0; 6.5]
**Waist circumference (cm) *p* = 0.004**					
Training	101.13 ± 10.82	100.82 ± 11.09	100.61 ± 9.04	98.59 ± 10.94 ^ab^	97.45 ± 11 ^abc^
Control	102.73 ± 10.25	103.02 ± 9.36	103.46 ± 9.43	104.24 ± 9.73 ^a^	103.64 ± 9.04
ES [95%CI]	−0.15 [−20.8; 20.5]	−0.22 [−20.3; 19.8]	−0.31 [−18.4; 17.8]	−0.84 [−21.1; 19.4]	−0.62 [−20.3; 19.0]
**Hip circumference (** **cm) *p* = 0.07**					
Training	106.67 ± 8.2	106.5 ± 8.37	106.43 ± 8.17	105.75 ± 8.49	105.76 ± 8.18
Control	105.34 ± 7.09	105.96 ± 6.77	106.49 ± 6.89 ^a^	106.65 ± 6.88 ^a^	106.55 ± 6.48
ES [95%CI]	0.17 [−14.8; 15.2]	0.07 [−14.7; 14.9]	−0.01 [−14.8; 14.8]	−0.12 [−15.2; 15.0]	−0.11 [−14.5; 14.3]
**Cardiovascular function**					
**SBP (mmHg) *p* = 0.73**	**W0**	**W4**	**W8**	**W12**	**W16**
Training	133.1 ± 14.7	133 ± 13.2	129.7 ± 11.1	125 ± 12.5	129.9 ± 14.7
Control	132 ± 22.7	129.9 ± 15.7	129 ± 13.9	127.9 ± 16.6	130.3 ± 14.4
ES [95%CI]	0.06 [−36.6; 36.7]	0.21 [−28.1; 28.5]	0.06 [−24.4; 24.6]	−0.2 [−28.7; 28.3]	−0.03 [−28.6; 28.5]
**DBP (mmHg) [Log] *p* = 0.69**					
Training	86 ± 8.2	84.3 ± 8.3	83.4 ± 6.3	81.9 ± 7.2	84.4 ± 10.2
Control	79.5 ± 9.5	80.3 ± 10	78.7 ± 9.3	79.7 ± 13.9	80 ± 10.3
ES [95%CI]	0.73 [−16.6; 18.1]	0.44 [−17.5; 18.4]	0.6 [−14.7; 15.9]	0.21 [−20.5; 20.9]	0.43 [19.7; 20.5]
**MAP (mmHg) *p* = 0.59**					
Training	101.7 ± 8.7	100.5 ± 8.9	98.8 ± 6.5	96.3 ± 7	99.6 ± 10.9
Control	97 ± 12.8	96.9 ± 10.5	95.5 ± 9.8	95.7 ± 13.5	96.8 ± 10.6
ES [95%CI]	0.44 [−20.6; 21.5]	0.37 [−18.6; 19.4]	0.4 [−15.6; 16.4]	0.06 [−20.0; 20.2]	0.26 [−20.8; 21.3]
**ABI [Log] *p* = 0.10**					
Training	1.15 ± 0.09	1.2 ± 0.1	1.22 ± 0.09	1.25 ± 0.09	1.26 ± 0.12
Control	1.17 ± 0.13	1.17 ± 0.09	1.2 ± 0.12	1.22 ± 0.1	1.18 ± 0.07
ES [95%CI]	−0.18 [−0.4; 0.0]	0.32 [0.1; 0.5]	0.19 [−0.0; 0.4]	0.32 [0.1; 0.5]	0.84 [0.7; 1.0]
**SV (ml) *p* = 0.88**					
Training	84 ± 15	96 ± 26	104 ± 30	96 ± 30	93 ± 34
Control	93 ± 31	107 ± 29	108 ± 26	107 ± 33	100 ± 31
ES [95%CI]	−0.39 [−45.5; 44.7]	−0.4 [−54.3; 53.5]	−0.14 [−55.0; 54.7]	−0.35 [−62.1; 61.4]	−0.22 [−63.9; 63.5]
**CO (L/min) *p* = 0.82**					
Training	5.57 ± 0.99	5.86 ± 1.47	6.35 ± 1.73	5.93 ± 1.73	5.54 ± 1.82
Control	6.14 ± 2.27	6.85 ± 2.2	7.01 ± 1.68	7.14 ± 2.07	6.63 ± 1.99
ES [95%CI]	−0.35 [−3.5; 2.9]	−0.54 [−4.1; 3.1]	−0.39 [−3.7; 3.0]	−0.64 [−4.4; 3.1]	−0.57 [−4.3; 3.2]
**TPR (mmHg/min/L) [Log] *p* = 0.64**					
Training	1.15 ± 0.31	1.11 ± 0.38	1.02 ± 0.29	1.09 ± 0.46	1.22 ± 0.55
Control	1.08 ± 0.34	0.93 ± 0.35	0.85 ± 0.24	0.84 ± 0.31	0.97 ± 0.36
ES [95%CI]	0.22 [−0.4; 0.9]	0.49 [−0.2; 1.2]	0.64 [0.1; 1.2]	0.65 [−0.1; 1.4]	0.55 [−0.3; 1.4]
**IMT (mm) *p* = 0.45**					
Training	0.048 ± 0.01	0.05 ± 0.01	0.047 ± 0.01	0.051 ± 0.01	0.052 ± 0.01
Control	0.046 ± 0.01	0.048 ± 0.01	0.046 ± 0.01	0.046 ± 0.01	0.046 ± 0.01
ES [95%CI]	0.2 [1.2; 0.2]	0.2 [0.2; 0.2]	0.1 [0.1; 0.1]	0.5 [0.5; 0.5]	0.6 [0.6; 0.6]
**CC (%/10 mmHg)** **[Box-Cox] *p* = 0.05**					
Training	1.05 ± 0.33	0.99 ± 0.44	1.03 ± 0.39	1.14 ± 0.48	0.84 ± 0.35
Control	0.87 ± 0.52	1.05 ± 0.42	0.96 ± 0.38	1.17 ± 0.51	1.24 ± 0.54
ES [95%CI]	0.42 [−0.4; 1.3]	−0.14 [−1.0; 0.7]	0.18 [−0.6; 0.9]	−0.06 [−1.0; 0.9]	−0.9 [−1.8; −0.0]
**Metabolic Syndrome Components (%)**					
**MetS (n)**	**W0 (23/23)**	**W4 (23/23)**	**W8 (23/23)**	**W12 (21/23)**	**W16 (21/23)**
Training	74	74	74	76	67 **
Control	87	91	91	91	91
**Waist circumference**					
Training	70	70	70	67	67
Control	78	78	83	78	83
**Hypertension**					
Training	91	91	91	90	95
Control	96	96	96	100	100
**Hyperglycemia**					
Training	22	30	22	19	24
Control	26	26	35	39	26
**High triglycerides**					
Training	35	43	43	38	38
Control	43	48	39	35	39
**Low HDL**					
Training	83	78	83	86	71
Control	87	87	91	87	83

**Legend:** Data are presented as mean ± SD. Mixed model analysis with group (Training and Control) and time (W0, W4, W8, W12 and W16) as fixed factors and subjects as random factors. n: number of participants analyzed in Training and Control, respectively; BMI: body mass index VL: vastus lateralis; RF: rectus femoris; SBP: systolic blood pressure; DBP: diastolic blood pressure; MAP: mean blood pressure; ABI: ankle-brachial index; SV: stroke volume; CO: cardiac output; TPR: total peripheral resistance; IMT: intima media thickness; CC: carotid compliance. [Log]: Analysis of log transformed data; [NN] Analysis of non-normally distributed data; [OR]: analysis with outliers removed; [Box–Cox]: Analysis of one of the Box–Cox family transformations. ^a^: different from W0; ^b^: different from W4; ^c^: different from W8; MetS: Metabolic Syndrome positive; ** different from Control on Chi squared between groups at each time point; ES: Effect Size (Cohen’s d); 95%CI: ninety five percent confidence interval. *p*-value: group*time interaction.

**Table 3 ijerph-19-11042-t003:** Physical fitness.

	W0	W8	W16
**Isokinetic peak torque of knee extension (Kg) [Log] *p* < 0.001**			
Training	118 ± 40.7	111.7 ± 40.7	126.7 ± 46.4 ^a^*
Control	114.6 ± 38	111 ± 36	102.9 ± 32.8 ^ac^
ES [95%CI]	0.09 [−77.0; 77.2]	0.02 [−75.2; 75.2]	0.6 [−77.0; 78.2]
**Isokinetic peak torque of knee flexion (Kg) [Log] *p* = 0.19**			
Training	58.9 ± 21.9	56.9 ± 20.5	67.2 ± 25.4
Control	58 ± 22.1	57.4 ± 21.1	72.4 ± 29.5
ES [95%CI]	0.04 [−43.1; 43.2]	−0.02 [−40.8; 40.7]	−0.19 [−54.0; 53.6]
**Isometric peak torque of knee extension (Kg) [Log] *p* = 0.007**			
Training	140.9 ± 52.1	139.7 ± 44.5	152.4 ± 54.4 *
Control	130.9 ± 38.1	136.3 ± 59.5	115.5 ± 34.9 ^c^
ES [95%CI]	0.22 [−88.2; 88.6]	0.07 [−101.9; 102.0]	0.83 [−86.7; 88.3]
**Isometric peak torque of knee flexion (Kg) [SqR] *p* = 0.06**			
Training	66.4 ± 27.6	71.8 ± 23.4	77.5 ± 27.1
Control	66.6 ± 27.1	69.6 ± 31.8	73.6 ± 27
ES [95%CI]	−0.01 [−53.6; 53.6]	0.08 [−54.0; 54.2]	0.14 [−52.9; 53.2]
**Rate of force development of knee extension [Log] *p* = 0.004**			
Training	0.055 ± 0.034	0.056 ± 0.039	0.074 ± 0.058
Control	0.071 ± 0.04	0.059 ± 0.031	0.053 ± 0.032 ^a^
ES [95%CI]	−0.43 [−0.5; −0.4]	−0.09 [−0.2; −0.02]	0.47 [0.4; 0.6]
**Rate of force development of knee flexion [Log] *p* = 0.05**			
Training	0.043 ± 0.043	0.059 ± 0.041	0.064 ± 0.051
Control	0.057 ± 0.044	0.039 ± 0.027	0.053 ± 0.042
ES [95%CI]	−0.32 [−0.4; −0.2]	0.59 [0.5; 0.7]	0.24 [0.2; 0.3]
$\dot{V}$ **O_2_max (ml/kg/min) *p* < 0.001**			
Training	18.5 ± 3.4 ^c^	20.2 ± 3.7 ^a^	21.9 ± 3.8 ^ac^*
Control	18.9 ± 3.9	19.3 ± 4.2	19.1 ± 3.6
ES [95%CI]	−0.32 [−0.4; −0.2]	0.23 [−7.5; 8.0]	0.76 [−6.5; 8.0]
**Maximal speed (km/h) *p* < 0.001**			
Training	7.9 ± 1 ^c^	8.3 ± 1.1 ^a^*	8.8 ± 1.2 ^ac^*
Control	7.8 ± 0.8	7.6 ± 0.6	7.6 ± 1
ES [95%CI]	0.11 [−1.6; 1.9]	0.82 [−0.8; 2.5]	1.09 [−1.1; 3.3]
**Balance (Berg Scale) [NN] *p* = 0.57**			
Training	54.8 ± 1.2	-	55.3 ± 0.9
Control	54.9 ± 1.2	-	55.2 ± 1.3
ES [95%CI]	−0.08 [−2.4; 2.3]	-	0.09 [−2.1; 2.3]
**Sit and reach (cm) *p* = 0.65**			
Training	19.8 ± 8.5	-	19.1 ± 7.6
Control	20.9 ± 12.3	-	20.2 ± 11.9
ES [95%CI]	−0.11 [−20.5; 20.3]	-	−0.11 [−19.2; 19.0]
**Stand and sit (repetitions in 30 s) [Log] *p* = 0.11**			
Training	11.8 ± 2.2	-	15 ± 3.1
Control	11.9 ± 3	-	13.6 ± 3.4
ES [95%CI]	−0.04 [−5.1; 5.1]	-	0.43 [−5.9; 6.8]
**Handgrip strength (Kg) [Log] *p* = 0.77**			
Training	30.2 ± 10.5	-	28.5 ± 9.8
Control	30.5 ± 10.2	-	28.9 ± 9.2
ES [95%CI]	−0.03 [−20.3; 20.3]	-	−0.04 [−18.7; 18.6]
**Timed up and go (s) *p* = 0.67**			
Training	8.2 ± 1.4	-	6.8 ± 1
Control	8.3 ± 1.1	-	7 ± 0.9
ES [95%CI]	−0.08 [−2.5; 2.4]	-	−0.21 [−2.1; 1.7]
**Gait speed (s) *p* = 0.46**			
Training	9.6 ± 1.4	-	7.8 ± 1.1
Control	9.5 ± 1	-	8 ± 0.9
ES [95%CI]	0.08 [−2.3; 2.4]	-	−0.2 [−2.2; 1.8]

**Legend:** Data are presented as mean ± SD. Mixed model analysis with group (Training and Control) and time (W0, W8 and W16) as fixed factors and subjects as random factors. ES: Effect size (Cohen’s d). 95%CI: ninety five percent confidence interval. [Log]: Analysis of log transformed data. [SqR]: Analysis of square root transformed data; [NN] Analysis of non-normally distributed data. ^a^: different from W0; ^c^: different from W8; *: different from Control. ES: Effect Size (Cohen’s d); 95%CI: ninety five percent confidence interval. *p*-value: group*time interaction.

**Table 4 ijerph-19-11042-t004:** Blood markers.

Glycemic and Lipidic Profile					
	**W0**	**W4**	**W8**	**W12**	**W16**
**Blood glucose (mg/dL) [Box-Cox] *p* = 0.03**					
Training	106.1 ± 22.1	105.5 ± 17.2	104.4 ± 15.6	102.9 ± 14.8	103.9 ± 21.1
Control	107.3 ± 16.1	106 ± 23.3	106 ± 16.4	109 ± 17.3	103 ± 17.5 ^d^
ES [95%CI]	−0.06 [−37.5; 37.4]	−0.02 [−39.7; 39.7]	−0.1 [−31.5; 31.3]	−0.38 [−31.8; 31.1]	0.05 [−37.8; 37.9]
**Insulin (μm/mL) [Log] *p* = 0.34**					
Training	4.5 ± 2.9	4.5 ± 2.6	4.3 ± 3.1	3.9 ± 2.7	5.2 ± 3.4
Control	5 ± 5.3	4.3 ± 4.3	4.9 ± 4.9	4.3 ± 4.1	4.5 ± 3.8
ES [95%CI]	−0.12 [−8.2; 7.9]	0.06 [−6.7; 6.8]	−0.15 [−8.0; 7.7]	−0.12 [−6.78; 6.6]	0.19 [−6.9; 7.3]
**HOMA-IR [NN] *p* = 0.36**					
Training	1.19 ± 0.86	1.19 ± 0.72	1.13 ± 0.88	0.99 ± 0.71	1.42 ± 1.14
Control	1.5 ± 1.85	1.33 ± 1.7	1.44 ± 1.64	1.3 ± 1.4	1.27 ± 1.22
ES [95%CI]	−0.23 [−2.9; 2.4]	−0.12 [−2.5; 2.3]	−0.25 [−2.7; 2.2]	−0.29 [−2.4; 1.8]	0.13 [−2.2; 2.4]
**Triglycerides (mg/dL) [Log] *p* = 0.38**					
Training	151.3 ± 168.5	140.2 ± 91.9	122.7 ± 65.9	123.1 ± 65	134 ± 81.7
Control	122.7 ± 56.1	114.2 ± 49.9	109.5 ± 40.5	109.4 ± 45	111.3 ± 49
ES [95%CI]	0.25 [−219.9; 220.4]	0.37 [−138.6; 139.3]	0.25 [−104.0; 104.5]	0.25 [−107.6; 108.0]	0.35 [−127.7; 128.4]
**Total cholesterol (mg/dL) [SqR] *p* = 0.55**					
Training	177.1 ± 42.4	178 ± 30.6	170.3 ± 39	170.1 ± 35.8	180 ± 48.2
Control	172.8 ± 42.1	174.4 ± 36.6	166.5 ± 38.7	161.6 ± 39.3	165 ± 41.8
ES [95%CI]	0.1 [−82.7; 82.9]	0.11 [−65.8; 66.0]	0.1 [−76.1; 76.2]	0.23 [−73.4; 73.8]	0.33 [−87.9; 88.5]
**HDL (mg/dL) [Log] *p* = 0.05**					
Training	40.3 ± 11.2	41.3 ± 15.1	39 ± 13.6	39.8 ± 15.3	43.7 ± 21
Control	39.5 ± 10.5	41.1 ± 10.5	39 ± 10.5	37.8 ± 12.1	38.3 ± 12.7
ES [95%CI]	0.07 [−21.2; 21.3]	0.02 [−25.1; 25.1]	0 [−23.6; 26.6]	0.15 [−26.7; 27.0]	0.32 [−32.7; 33.4]
**LDL (mg/dL) *p* = 0.83**					
Training	107.8 ± 29.2	109.7 ± 23.9	106.8 ± 30.1	105.7 ± 25.6	108.7 ± 31.4
Control	108.8 ± 34.7	110.4 ± 32.1	105.7 ± 33.3	102 ± 33.8	104.3 ± 33.3
ES [95%CI]	−0.03 [−62.7; 62.6]	−0.03 [−54.9; 54.9]	0.03 [−62.1; 62.2]	0.12 [−58.1; 58.3]	0.14 [−63.3; 63.5]
**Inflammatory profile**					
	**W0**	**W4**	**W8**	**W12**	**W16**
**IL-6 (pg/mL) [Box-Cox] *p* = 0.29**					
Training	3.92 ± 5.64	3.95 ± 6.06	3.56 ± 4.39	2.8 ± 3.06	3.44 ± 4.85
Control	3.53 ± 3.23	2.92 ± 1.77	3.25 ± 1.89	4.41 ± 5.22	3.47 ± 2.56
ES [95%CI]	0.09 [−8.6; 8.8]	0.26 [−7.4; 7.9]	0.1 [−6.1; 6.3]	−0.39 [−8.5; 7.8]	−0.01 [−7.3; 7.3]
**IL-1ra (pg/mL) [Log] *p* = 0.40**					
Training	840 ± 568	889 ± 614	998 ± 866	1013 ± 869	987 ± 755
Control	673 ± 308	640 ± 284	632 ± 275	737 ± 544	716 ± 383
ES [95%CI]	0.38 [−858.1; 858.9]	0.55 [−879.5; 880.6]	0.64 [−1117.5; 1118.8]	0.39 [−1384.4; 1358.1]	0.48 [−1114.8; 1115.7]
**TNF-α (pg/mL) *p* = 0.90**					
Training	1.58 ± 0.36	1.59 ± 0.35	1.63 ± 0.41	1.75 ± 0.45	1.73 ± 0.52
Control	1.53 ± 0.48	1.39 ± 0.48	1.44 ± 0.49	1.49 ± 0.56	1.64 ± 0.52
ES [95%CI]	0.12 [−0.7; 0.9]	0.48 [−0.3; 1.3]	0.42 [−0.5; 1.3]	0.51 [−0.5; 1.5]	0.17 [−0.9; 1.2]
**PCR (mg/L) [Log] *p* = 0.83**					
Training	1.4 ± 1.2	1.8 ± 1.5	1.7 ± 1.6	1.4 ± 1	1.2 ± 0.7
Control	1.8 ± 1.3	1.6 ± 1.4	1.6 ± 1.7	1.7 ± 1.5	1.5 ± 1.3
ES [95%CI]	−0.32 [−2.8; 2.13]	0.14 [−2.7; 3.0]	0.06 [−3.2; 3.3]	−0.24 [−2.7; 2.2]	−0.3 [−2.3; 1.7]
**Adiponectin (μg/mL) [Log] *p* = 0.91**					
Training	4.5 ± 1.5	4.5 ± 1.7	4.4 ± 1.7	4.2 ± 1.6	4.4 ± 1.7
Control	5.2 ± 2.6	5.4 ± 3	5.3 ± 2.7	5 ± 2.7	5.4 ± 2.8
ES [95%CI]	−0.34 [−4.4; 3.7]	−0.38 [−5.0; 4.2]	−0.41 [−4.7; 3.9]	−0.37 [−4.6; 3.8]	−0.44 [−4.9; 4.0]
**Leptin (ng/mL) [Log] *p* = 0.08**					
Training	49.9 ± 52.2	53.5 ± 52.9	48.4 ± 49.2	48.5 ± 50.3	46.5 ± 49.5
Control	40.9 ± 29	41 ± 30.4	43.2 ± 32	44.4 ± 29.4	41 ± 25.1
ES [95%CI]	0.22 [−79.4; 79.8]	0.3 [−81.3; 81.9]	0.13 [−79.5; 79.7]	0.1 [−78.0; 78.2]	0.15 [−73.0; 73.3]
**Oxidative stress profile**					
	**W0**	**W4**	**W8**	**W12**	**W16**
**Carbonils (nmol/mg) [Log] *p* = 0.39**					
Training	1.35 ± 0.33	1.35 ± 0.27	1.39 ± 0.34	1.55 ± 0.3	1.46 ± 0.27
Control	1.32 ± 0.36	1.35 ± 0.37	1.25 ± 0.35	1.34 ± 0.31	1.38 ± 0.37
ES [95%CI]	0.09 [−0.6; 0.8]	0 [−0.6; 0.6]	0.41 [−0.3; 1.1]	0.69 [0.1; 1.3]	0.25 [−0.4; 0.8]
**TBARS (µmoles/mg) [OR NN] *p* = 0.28**					
Training	3.98 ± 0.99	4.14 ± 1.16	4.11 ± 0.9	4.4 ± 0.96	4.76 ± 1.57
Control	4.86 ± 3.27	3.92 ± 1.08	4.33 ± 1.02	4.83 ± 1.66	4.39 ± 1.05
ES [95%CI]	−0.41 [−4.6; 3.8]	0.2 [−2.0; 2.4]	0.81 [−1.1; 2.7]	−0.33 [−2.9; 2.2]	0.28 [−2.3; 2.9]
**Hydrogen peroxide (µM) [Box-Cox] *p* = 0.14**					
Training	151 ± 307.1	90.4 ± 70.2	70.3 ± 34.9	78.7 ± 43.6	106.4 ± 101.5
Control	97.1 ± 63.8	75 ± 42.4	104.1 ± 130.5	76.4 ± 31.1	92.9 ± 55
ES [95%CI]	0.29 [−363.2; 363.7]	0.27 [−110.1; 110.6]	0.41 [−162.5; 161.7]	0.06 [−73.1; 73.3]	0.17 [−153.5; 153.5]
**NADPH oxidase (ng/mg) [SqR] *p* = 0.30**					
Training	0.017 ± 0.094	0.033 ± 0.022	0.041 ± 0.032	0.042 ± 0.027	0.026 ± 0.028
Control	0.037 ± 0.02	0.034 ± 0.023	0.034 ± 0.033	0.029 ± 0.028	0.046 ± 0.049
ES [95%CI]	−0.35 [−0.5; −0.2]	−0.04 [−0.09; 0]	0.22 [0.2; 0.3]	0.47 [0.4; 0.5]	−0.52 [−0.6; −0.4]
**FRAP (mM Fe(ii) [Log] *p* = 0.66**					
Training	1.44 ± 0.4	1.4 ± 0.27	1.33 ± 0.29	1.47 ± 0.43	1.42 ± 0.48
Control	1.33 ± 0.37	1.32 ± 0.37	1.34 ± 0.38	1.29 ± 0.4	1.31 ± 0.42
ES [95%CI]	0.29 [−0.5; 1.0]	0.25 [−0.4; 0.9]	−0.03 [0.7; 0.6]	0.43 [−0.4; 1.3]	0.24 [−0.6; 1.1]
**SOD (Usod/mg protein) [Log] *p* = 0.17**					
Training	3.82 ± 0.62	3.89 ± 0.58	3.96 ± 0.6	3.98 ± 0.72	3.81 ± 0.62
Control	3.85 ± 0.46	3.91 ± 0.63	3.74 ± 0.76	4.06 ± 0.76	3.95 ± 0.65
ES [95%CI]	−0.06 [−1.1; 1]	−0.03 [−1.2; 1.2]	0.32 [−1.0; 1.7]	−0.11 [−1.6; 1.3]	−0.22 [−1.5; 1.0]
**Vasoactive substances**					
	**W0**	**W4**	**W8**	**W12**	**W16**
**ET-1 (pg/mL) [SqR] *p* = 0.23**					
Training	9.2 ± 1.4	9.1 ± 1.3	9 ± 1.2	9.2 ± 1.5	9.3 ± 1.5
Control	9.2 ± 1.3	8.9 ± 1.3	9 ± 1.1	9.2 ± 1.5	8.9 ± 1.2
ES [95%CI]	0 [−2.7; 2.7]	0.15 [−2.4; 2.7]	0 [−2.3; 2.3]	0 [−2.9; 2.9]	0.3 [−2.4; 2.9]
**Nitrite (nmol/mg protein) [Log] *p* = 0.96**					
Training	0.634 ± 1.011	0.481 ± 0.548	0.443 ± 0.588	0.518 ± 0.524	0.54 ± 0.489
Control	0.554 ± 0.463	0.565 ± 0.806	0.399 ± 0.291	0.581 ± 0.604	0.499 ± 0.435
ES [95%CI]	0.11 [−1.3; 1.6]	−0.12 [−1.5; 1.2]	0.1 [−0.8; 1.0]	−0.11 [−1.2; 1.0]	0.09 [−0.8; 1.0]

**Legend:** Data are presented as mean ± SD. [Log] Analysis of log transformed data. [SqR]: Analysis of square root transformed data; [NN] Analysis of non-normally distributed data; [OR]: analysis with outliers removed; [Box–Cox]: Analysis of one of the Box–Cox family transformations; HDL: High density lipoprotein; LDL: Low density lipoprotein; IL-1ra: interleukin 1 receptor antagonist; IL-6: interleukin 6; TNF-α: Tumor necrosis factor alpha; CRP: C-reactive protein; TBARS: Thiobarbituric acid reactive substances; NASDH: nicotinamide adenine dinucleotide phosphate oxidase; FRAP: Ferric Reduction Ability Power; SOD: superoxide dismutase; ET-1: endothelin-1; ES: Effect Size (Cohen’s d); 95%CI: ninety five percent confidence interval; *p*-value: group*time interaction; ^d^: different from W12.

**Table 5 ijerph-19-11042-t005:** Heart rate variability indexes.

Time Domain					
**FC (bpm) *p* = 0.06**	**W0**	**W4**	**W8**	**W12**	**W16**
Training	61 ± 8	61 ± 7	61 ± 7	61 ± 7	60 ± 7
Control	66 ± 8	65 ± 7	66 ± 7	67 ± 7	66 ± 8
ES [95%CI]	−0.63 [−16.3; 15.1]	−0.57 [−14.3; 13.2]	−0.71 [−14.4; 13.0]	0 [−13.7; 13.7]	−0.8 [−15.5; 13.9]
**RRi (ms) [Log] *p* = 0.48**					
Training	995 ± 139	990 ± 117	986 ± 110	993 ± 122	1015 ± 113
Control	916 ± 114	938 ± 96	920 ± 98	912 ± 94	921 ± 104
ES [95%CI]	0.62 [−247.3; 248.6]	0.49 [−208.3; 209.2]	0.63 [−203.2; 204.5]	0.75 [−210.9; 212.4]	0.87 [−211.8; 213.5]
**SDNN (ms) [Box-Cox] *p* = 0.20**					
Training	29 ± 12	31 ± 14	32 ± 15	28 ± 11	35 ± 18
Control	28 ± 14	32 ± 12	30 ± 16	33 ± 16	29 ± 15
ES [95%CI]	0.08 [−25.4; 25.6]	−0.08 [−25.6; 25.4]	0.13 [−30.3; 30.5]	−0.37 [−26.8; 26.1]	0.36 [−32.0; 32.7]
**RMSSD (ms) [Log] *p* = 0.38**					
Training	22.4 ± 9.2	23.4 ± 11.5	25.5 ± 14.4	23 ± 11.1	26 ± 16.4
Control	21.8 ± 13.9	27.1 ± 14.8	23.6 ± 15.3	26.2 ± 17.1	23.2 ± 14.8
ES [95%CI]	0.05 [−22.6; 22.7]	−0.28 [−26.1; 25.5]	0.13 [−29.0; 29.2]	−0.23 [−27.9; 27.4]	0.18 [−30.4; 30.8]
**pNN50 (%) [Log] *p* = 0.87**					
Training	4.7 ± 7	5.6 ± 11	7.9 ± 12.3	5 ± 8.3	7.9 ± 13.6
Control	6.7 ± 9.8	10 ± 11.8	7.8 ± 13.2	10 ± 14.7	7.9 ± 12.6
ES [95%CI]	−0.24 [−16.7; 16.2]	−0.39 [−22.7; 22.0]	0.01 [−25.0; 25.0]	−0.43 [−23.0; 22.1]	0 [−25.7; 25.7]
**Frequency domain**					
**HF (ms^2^) [Log] *p* = 0.52**	**W0**	**W4**	**W8**	**W12**	**W16**
Training	163 ± 104	216 ± 275	270 ± 335	231 ± 247	333 ± 530
Control	277 ± 422	294 ± 315	258 ± 329	274 ± 339	293 ± 386
ES [95%CI]	−0.43 [−515.9; 515.1]	−0.26 [−578.5; 577.9]	0.04 [−650.7; 650.8]	−0.15 [−574.4; 574.1]	0.09 [−897.6; 897.8]
**LF (ms^2^) [Log] *p* = 0.23**					
Training	243 ± 370	239 ± 268	284 ± 377	221 ± 230	306 ± 383
Control	196 ± 255	283 ± 300	254 ± 381	307 ± 352	223 ± 306
ES [95%CI]	0.15 [−612.4; 612.7]	−0.15 [−556.8; 556.5]	0.08 [−742.8; 742.9]	−0.3 [−570.7; 570.1]	0.24 [−675.0; 675.5]
**TP [NN] *p* = 0.42**					
Training	928 ± 1487	1008 ± 987	1156 ± 1396	853 ± 733	1220 ± 1123
Control	895 ± 1019	1047 ± 783	1237 ± 1627	1257 ± 1307	887 ± 1204
ES [95%CI]	0.03 [−2455.9; 2455.9]	−0.04 [−1734.6; 1734.6]	−0.05 [−2962.6; 2962.5]	−0.4 [−1999.6; 1999.8]	0.29 [−2280.2; 2280.8]
**LF/HF [Log] *p* = 0.74**					
Training	1.5 ± 1.17	2.13 ± 2.65	1.76 ± 2.38	2.11 ± 3.65	1.7 ± 1.66
Control	1.68 ± 1.51	1.6 ± 1.33	1.71 ± 1.59	1.76 ± 1.86	0.58 ± 0.51
ES [95%CI]	−0.13 [−2.8; 2.5]	0.27 [−3.6; 4.2]	0.03 [−3.9; 3.9]	0.13 [−5.3; 5.5]	1.03 [−1.1; 3.2]
**HF (nu) *p* = 0.90**					
Training	47 ± 17.5	45.2 ± 21.5	47.5 ± 19.7	48.6 ± 22	46.3 ± 19.1
Control	48 ± 23	46.7 ± 18.3	46.7 ± 20.3	44.4 ± 15.8	49 ± 19.3
ES [95%CI]	−0.05 [−39.7; 39.6]	−0.08 [−39.1; 38.9]	0.04 [−39.2; 39.2]	0.22 [−36.8; 37.3]	−0.14 [−37.8; 37.5]
**LF (nu) *p* = 0.87**					
Training	52.6 ± 17.7	54.7 ± 21.6	52.4 ± 19.8	51.3 ± 22.1	53.6 ± 19.2
Control	51.9 ± 23.1	53.1 ± 18.3	53.2 ± 20.4	55.2 ± 15.9	49.7 ± 18.7
ES [95%CI]	0.03 [−40.0; 40.0]	0.08 [−39.0; 39.2]	−0.04 [−39.4; 39.4]	−0.21 [−37.5; 37.0]	0.21 [−36.9; 37.4]
**Non-linear analysis.**					
**SD1 [Log] *p* = 0.77**	**W0**	**W4**	**W8**	**W12**	**W16**
Training	15.9 ± 6.5	16.6 ± 8.1	18.1 ± 10.2	16.3 ± 7.9	18.4 ± 11.6
Control	15.5 ± 9.8	18.7 ± 10.9	16.7 ± 10.8	18.6 ± 12.1	17.9 ± 13.9
ES [95%CI]	0.05 [−15.9; 16.0]	−0.22 [−18.8; 18.4]	0.13 [−20.5; 20.7]	−0.23 [−19.8; 19.4]	0.04 [−25.0; 25.0]
**SD2 [NN] *p* = 0.02**					
Training	37.2 ± 17.4	40.4 ± 19.1	41.5 ± 20.8	36.5 ± 13.5	45.5 ± 23.1
Control	36.1 ± 17	41.2 ± 15	38.8 ± 20.2	42.9 ± 19.3	34.1 ± 19.5
ES [95%CI]	0.06 [−33.7; 33.8]	−0.05 [−33.5; 33.4]	0.13 [−40.0; 40.3]	−0.39 [−32.5; 31.8]	0.54 [−41.2; 42.3]

**Legend:** Data are presented as mean ± SD. Mixed model analysis with group (Training and Control) and time (W0, W4, W8, W12 and W16) as fixed factors and subjects as random factors. HR: heart rate; RRi: R–R interval; SDNN: standard deviation of all normal R–R intervals; RMSSD: square root of the mean squared differences of successive R–R intervals; pNN50: Percentage of successive RR intervals that differ by more than 50 ms; HF: high frequency; LF: low frequency; VLF: very low frequency; TP: total power; LF/HF: ratio between the bands of low and high frequencies; SD1: Poincaré plot standard deviation perpendicular along the line of identity; SD2: Poincaré plot standard deviation along the line of identity; [Log] Analysis of log transformed data. [SqR]: Analysis of square root transformed data; [NN] Analysis of non-normally distributed data; [Box–Cox]: Analysis of one of the Box–Cox family transformations. ES: Effect Size (Cohen’s d); 95%CI: ninety five percent confidence interval. *p*-value: group*time interaction.

## Data Availability

The data presented in this study are available on request from the corresponding author.

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
