# Peer review of "Comprehensive Time-Course Effects of Combined Training on Hypertensive Older Adults: A Randomized Control Trial"

_ijerph, 2022, doi:10.3390/ijerph191711042_

Round 1

Reviewer 1 Report

Please see attached file, the parts marked in yellow

Thank you

Author Response

We thank you one more time to evaluate with attention our manuscript. We have revised it with all changes marked in yellow and our detailed response to each comment is described below.

We are looking forward to your response.

Yours sincerely

  • It seems that the authors have tried only the Log and Sqr transformations and retain the transformation which made the data look normally distributed. It would be interesting to try the Box-Cox family in the non-normal variables and find out whether another transformation fulfills the normality objective.

Authors’ response: Thank you for your suggestion. We applied the transformation suggested for all the variables not normally disributed by our first transformation methods and when they became normally distributed by one of the box-cox families we run the MMA model again with the new dataset. The only new result was a significant reduction of blood glucose in the control group from W12 to W16; and all the new P-values for the normalized data were replaced eventhough they were mantained non-significant.

  • The reply does not really address the original comment (i.e.the use of GLMs instead of transformations, at least for the variables where a transformation is not working adequately), but instead raises new points. It seems that heterogeneity within groups prevents any statistical significance across groups. In the models employed such heterogeneity could be modeled explicitly, for example by allowing different variance components for the two groups. In any case, since this line of reasoning goes far away, the reviwers accepts author`s reply, but still there is significant room for improvement in the analysis dataset.

Authors’ response: Thank you for understanding, we tryied the new transformations to seek for more improvement and since the results were reproducible one more time with the analysis of the normally distributed dataset we believe it reiforce the robustness of these findings.

  • Comment accepted, but still a simpler model in stochastic terms may provide a better fit. Since consecutive observations were measured 4 weeks apart, its hard to believe that an AR(1) structure is needed. Selection criteria are a useful tool in 3 deciding on the final covariance structure of the data.

Authors’ response: Thank you.

  • OK, but it sounds more likely that authors targeted a common model across all response variables, without exploring adequately the effect of other covariates. In any case if no covariates were significantly associated with any response then authors can leave things as they are.

Authors’ response: Thank you.

  • OK, but since the interaction is not important (i.e the curves for the two groups are statistically parallel), with the exception of a few cases, then showing these p-values does not add anything interesting. On the contrary p-values displaying important differences at specific time points between the two groups are more valuable. Also effects should be presented on the original and not transformed scale. Finally due to the limited sample, a significance level of 10% might be used.

Authors’ response: Thank you for new suggestions. When we have multiple comparisons in the model like 2 groups and 5 time points we may look to post-hoc tests only when we see significant  interactions. In any case, since we were so desperate to find significant results, we have been looked for post hoc significant P-values even when there was no interactions, but we really did not find more than what we alreday reported. The P-value of 10% will fit within what we called trend to be significant and we usually comment about it reinforcing the limitation of its use. However, even looking for these trends we could not find more than was already reported. In addition, we looked again for the variables with a significance level of 10% for interaction and we checked the post-hocs to report in the table, but there was no new significant effect there to be reported eventhough we were considering the 10% level.

  • If in the above graphical display, the statistically significant effects were added at each time point, then it would have been an excellent tool for communicating the study findings. See also previous comment, concentrating on the lime points where important results occurred.

Authors’ response: We are not sure if we understand this comment cause we have each time point in the figure and only the signifcant changes were reported.

Reviewer 2 Report

Dear authors,

Thank you very much for your work. However, I do have some questions. Please find my specific comments below:

Introduction

Comment 1.   You use the Microsoft Word template to prepare the manuscript, but it would be better if you could increase the line number. (https://www.mdpi.com/journal/ijerph/instructions)

Methods

Comment 1.   In the introduction of subjects, you reported that in order to ensure proper balance between the control group and the intervention group, you continuously inverted a few participants’ matched pair until the baseline difference between groups reached an effect size smaller than 0.2 (Cohen`s d) for each of the five primary variables. But in Table 1, you only report that the SE of age, body, SBP and PAL are less than 0.2. I suggest you check it again. It would be better if reporting could be added, specifically referring to the five primary variables.

Comment 2.  In the introduction of intervention, you mentioned the use of 15-minute strength training twice a week. Is there any arrangement for subjects to warm up before strength training to prevent injury during exercise? If there is a warm-up arrangement, how to operate it. 

Comment 3.  In Table 1, you reported the data of pal, but I didn't find the corresponding scale.

PAL: Physical activity level during the past week by a Likert type scale based on responses to 16 items (61); 

Comment 4.   Similarly, what does (60)(61) mean in Table 1? 

Comment 5.   In the outcomes section, I didn't find the physical fitness measurement.

Results

Comment 1.   Figure 1 is a little bit unclear. It would be better if you could download the Flow Diagram (http://www.consort-statement.org/), modify it on this basis, and then replace it. 

Comment 2.   The tables are informative. I do not see how to reduce them. They are a bit overwhelming. 

Comment 3.   For example, from table 5, I can't clearly find out which variables have groups*time interaction effect. 

Comment 4.   In the results section, you reported that 23 people in each group were included in the analysis (Figure 1), but in your supplementary Table S1, the results of ANOVA were mostly less than 46 people, such as 39 people in Maximal speed test, 40 people in sit and reach test, etc. What is the reason for the loss of data?

Comment 5.  From Table 3 you can remove the p-value and instead use **. 

Comment 6.  It would be better if you could replace the data in the table with a statistical chart.

Discussion

Comment 1.   In your research, I didn't find a specific time to carry out the experiment. Have you ever considered the correlation between hypertension and seasonal changes and meteorological factors? As far as I know, there have been many studies on the correlation between hypertension and seasonal changes in the elderly.

Reference

Comment 1.   Your references may be in any style, provided that you use the consistent formatting throughout. In your references, some have volume numbers and some do not.

[4],[5] are missing volume.

[10], [11], [18], [22], [23], [26], [30] and [37] are missing volume number and page number.

It is suggested to modify according to the following format:

 Journal reference: Bowman, C.M.; Landee, F.A.; Reslock, M.A. Chemically Oriented Storage and Retrieval System. 1. Storage and Verification of Structural Information. J. Chem. Doc. 1967, 7, 43-47; DOI:10.1021/c160024a013.

References to books should cite the author(s), title, publisher, publisher location (city and country), publication year, and page:

9. Smith, A.B. Textbook of Organic Chemistry; D. C. Jones: New York, NY, USA, 1961; pp 123-126.

Comment 2.   It is suggested to check carefully and modify the format according to the requirements of MDPI for references (https://www.mdpi.com/authors/references).

Author Response

We thank you one more time to evaluate with attention our manuscript. We have revised it with all changes marked in yellow and our detailed response to each comment is described below.

We are looking forward to your response.

Yours sincerely

Introduction

Comment 1. You use the Microsoft Word template to prepare the manuscript, but it would be better if you could increase the line number. (https://www.mdpi.com/journal/ijerph/instructions)

Authors’ response: We noticed some parts of the text had 1.15 space between lines instead of 1.5, so we checked everything to make sure they were 1.5 now. We were not sure whether we understood your suggestion, so please ask for correction again if we did not meet the expectations. In any case, I believe the journal may edit this in case the manuscript is approved for publication.

Methods

Comment 1. In the introduction of subjects, you reported that in order to ensure proper balance between the control group and the intervention group, you continuously inverted a few participants’ matched pair until the baseline difference between groups reached an effect size smaller than 0.2 (Cohen`s d) for each of the five primary variables. But in Table 1, you only report that the SE of age, body, SBP and PAL are less than 0.2. I suggest you check it again. It would be better if reporting could be added, specifically referring to the five primary variables.

Authors’ response: Yes, we forgot to add the sex percentage there, and now we added it.

The other values are correct because we balanced the groups in the begining of the study, after the first assessments of all individuals and we lost some participants along follow-up, as it was reported in figure 1. So table 1 shows the ES difference after the end of the study, only for the participants that completed the study and thus we lost the balance for Rri and BMI. To clarify this information we added a sentence in the begining of results section.

Comment 2. In the introduction of intervention, you mentioned the use of 15-minute strength training twice a week. Is there any arrangement for subjects to warm up before strength training to prevent injury during exercise? If there is a warm-up arrangement, how to operate it.

Authors’ response: No, we did not instruct any specific type of warm-up. The total load of strength training was very easy to handle by them, since the volume of sets and repetitions were very low and usually they reach the gym by at least 10 min walk.

Comment 3. In Table 1, you reported the data of pal, but I didn't find the corresponding scale.

PAL: Physical activity level during the past week by a Likert type scale based on responses to 16 items (61);

Authors’ response: The final results of PAL is displayed in arbritary units and not a likert type scale, so we though it would be better to remove it from the table. We added a line to comment that despite we selected non-physically active participants we did assess it at baseline, but it was just for carachterization porpose and the value is a result of complex equation that includes the intensity of their activities (assessed by likert type scale) and the duration of those activities.

Comment 4. Similarly, what does (60)(61) mean in Table 1?

Authors’ response: We apologize for this mistake. It was a citation that we had in the previous version of the paper in which we had a very long methods description. Now we removed it from there. Since we noticed the reference for that was not in the text, we added it in the methods.

Comment 5. In the outcomes section, I didn't find the physical fitness measurement.

Authors’ response: We understand it was not clear, because we called physcial fitness in the results the combination of the following: cardiorespiratory fitness, strength and functionality tests. These assessments were described in a section called “Maximum oxygen consumption ( O2max), maximum strength and functionality” and to improve it we adapt the title to: Physical fitness (Maximum oxygen consumption ( O2max), maximum strength and functionality). The description of functionality was brief since we used very standardized tests and we did not change from the original protocol proposed in our methods paper (reference 10).

Round 2

Reviewer 1 Report

Please find attached comments on the 2nd revision marked in green

Thank you

Author Response

We replied to the comments in blue, since we understand the reviewers want to keep all the previous discussion close to each other in the same file.

Yours sincerely

Reviewer 1:

Statistical Review

  1. The authors state that two kinds of transformations were tried to achieve normality in the outcome variables: the logarithm (Log) and the square root (SqR). Both of them are special cases of a larger family of transformations, titled the Box-Cox family. For positive variables, this family is indexed by one parameter (usually called lambda in the literature), and estimation of this parameter reveals which transformation serves our purpose best. Typical examples of the Box-Cox family, apart from the Log and SqR transformations, are the Reciprocal and power transformations with integer values.

Author’s reply: Thank you for all these detailed information. We tested these two transformations and the transformations we maintained in the final analysis (reported in the table) were the ones that made the data become normally distributed. We described it on methods section.

Reviewer’s comment: It seems that the authors have tried only the Log and Sqr transformations and retain the transformation which made the data look normally distributed. It would be interesting to try the Box-Cox family in the non-normal variables and find out whether another transformation fulfills the normality objective.

Reviewer’s comment on V2: The authors have measured a wealth of clinically important variables. Replacing the “NN” tag with “Box-Cox” in some summary tables, it is not what has been suggested. The point is to run the Box-Cox procedure on all NN variables and decide whether there is a transformation which make the data look more normally distributed. The point has not been addressed. 

Author’s reply 2: We believe there is a misunderstanding here.The detailed procedures were described on methods section. We applied the Box-Cox transformations in all non-normally distributed variables. The mean and standard deviation did not change in the tables because we reported absolute results for all variables, even when they were transformed for analysis, because transformed values are not intuitive for interpretation, and it is not a common practice to report mean and SD of transformed values. We named Box-Cox on table, only the data that were analyzed by Box-cox transformed data and we only analyzed it when the transformation worked. If we applied all our transformation tools, including Box-cox and the data did not become normal, we just run analysis with non-normalized data and that was what we called NN.

To facilitate the reviewers corrections here is the part of the text where we described it:

The non-normally distributed data were transformed by logarithm ([Log]), or square root ([SqR]) transformations, or the whole Box-Cox family ([Box-Cox]). All data were described in the text and tables in raw format as mean ± standard deviation; additionally, we tagged the variables as [Log], [SqR] or no tag (when raw data was already normally distributed), according to their required transformation to become normally distributed for analysis. When none of the transformations led to normal distribution, we analyzed the raw format and target it as [NN] (non-normally distributed).

  1. Transforming to normality is not an end in itself. Today an analyst can fit a wide range of skewed or multi-modal distributions, which on the one hand they follow closely the histogram of the observed data and on the other hand they tackle appropriately the issue of outliers. Good examples are the Gamma or t distribution, or other densities within the GLM family. These models also allow the inclusion of random subject effects or other more complicated covariance structures for the repeated measurements of the response.

Author’s reply: Thank you again for the detailed information. Although we are not statisticians, we dedicated a lot to study and discuss with statistician all our analysis and we tested many models before we opted for the MMA as we did. I guess it is important to clarify to you that we were very disappointed with the absence of significance for many healthy markers that previous studies with even less power have shown and since we finish this study, we have been presenting these data in conference and discussed with many experts in the field that were intrigued with our findings. Thus, we tested simpler models including comparisons between groups for only one time-point. We needed to look individual by individual to understand why we could not see significance in any of the statistics. With these observations we obtained 2 important information: 1) each group of data was very heterogeneous, and that is why we opted to show individual data (Supplementary material), and add the effect sizes with 95% confidence interval to the mean and standard deviation information. 2) in the supplementary material, the individual data make it very clear that we did not find significant changes because the data across the 5 time-points were very stable (also reinforcing the quality of our controlled intervention and data collection). Thus, we confirmed by a comprehensive exploration/observation of the data (not only statistical analysis) that participants in both groups did not undergo visible changes besides their physical fitness and a small improvement in body composition with training. To avoid the type 2 error in the study and transparently confirm what we said before, we add a table in the supplementary material with the analysis of ANOVA repeated measures only for time points W0 and W16. In this way we apply the same design we used to calculate the sample size/power of the study.

Reviewer’s comment: The reply does not really address the original comment (i.e.the use of GLMs instead of transformations, at least for the variables where a transformation is not working adequately), but instead raises new points. It seems that heterogeneity within groups prevents any statistical significance across groups. In the models employed such heterogeneity could be modeled explicitly, for example by allowing different variance components for the two groups. In any case, since this line of reasoning goes far away, the reviewers accepts author’s reply, but still there is significant room for improvement in the analysis of this dataset.

Reviewer’s comment on V2: Not addressed. This point can be seen as an alternative to Box-Cox for the NN variables.

Author’s reply 2: We understand that this comment is related to the previous one and we reinforce that we applied Box-cox transformations in all non-normally distributed variables, leaving out just 5 NN for analysis (named: balance, HOMA-IR, TBARS, TP and SD2) since none of the transformations worked for them. In fact, there is heterogeneity within groups, but the mixed model we applied for analysis considered subjects as random factors and if there was some standard in the changes across time it would have been identified by the model.

3. The authors mention that both a participants random effect was included in the model, as well as a first order auto-regressive structure for the error term. This sounds like a heavy model regarding its stochastic terms. More likely to fit an independent, a compound symmetry or an AR(1) error structure and choose between them based by utilizing some well-known model selection criteria (e.g. BIC). There are well established protocols for selecting terms in the systematic part of a model after deciding on the stochastic part.

Author’s reply: We agree that this is a very heavy model but we opted to show it after the simple analysis we previously mentioned.

Reviewer’s comment: Comment accepted, but still a simpler model in stochastic terms may provide a better fit. Since consecutive observations were measured 4 weeks apart, its hard to believe that an AR(1) structure is needed. Selection criteria are a useful tool in deciding on the final covariance structure of the data.

Reviewer’s comment on V2: Not addressed, still the same model is applied

Author’s reply 2: The variations across time were very predictable and not stochastic, since each participant maintained very stable assessments across time. Thus the variation in our model comes from differences between participants and that is why we opted for AR(1) model. In the beginning of our exploration of data we did identify strong correlations across time points, and it reinforced the used of this model as it would be expected by our study design. In any case, we also tested the models suggested by the reviewers in the first round of review as presented on supplementary material and it did not lead to different results, that is why we opted to maintain the main results presented with more robust statistic.

4. It seems that in the initial model, the only terms considered are group, time and the interaction group by time, alongside with the random participant effect. Baseline characteristics (see Table 1) should also be considered.

Author’s reply: we tested analysis using baseline characteristics as covariable at the beginning of our exploratory analysis for many variables that were near significance, however as we were obtained same results than only include subjects as random effects, we opted to keep only subjects as random effects in the mixed model.

Reviewer’s comment: OK, but it sounds more likely that authors targeted a common model across all response variables, without exploring adequately the effect of other covariates. In any case if no covariates were significantly associated with any response then authors can leave things as they are.

Reviewer’s comment on V2: Not addressed. No information is presented that the effect of baseline characteristics on the various outcomes has been formally tested and non-significant results occur.

Author’s reply 2: We just tested the effects of covariates in the very beginning, and as you mentioned previously, the main covariates that could influence the results were baseline differences, such as sex or the own baseline values and these effects were corrected by the use of subjects as random effects in the mixed models.

  1. P-values should be added throughout the tables

Author’s reply: Thank you for the suggestion, we added the P-values for interaction in the tables.

Reviewer’s comment: OK, but since the interaction is not important (i.e the curves for the two groups are statistically parallel), with the exception of a few cases, then showing these p-values does not add anything interesting. On the contrary p-values displaying important differences at specific time points between the two groups are more valuable. Also effects should be presented on the original and not transformed scale. Finally due to the limited sample, a significance level of 10% might be used.

Reviewer’s comment on V2: Not addressed, still the same information is presented

Author’s reply 2: We changed our significance level to 10% as the reviewer requested. The new significant variables were: Hip circumference, CC, Isometric peak torque of knee flexion, and Rate of force development of knee flexion. Although there was interaction for all of them with an alpha value of 10%, only hip circumference had significant post-hoc showing increase of hip circumference at 8W and 12W compared to baseline for control group. The reason why the same information is presented is because, we already inserted on tables significant post hocs (previously at an alpha of 5%) for the tendency to interactions (alpha of 10%) even though we did not discuss it considering a significant change.

6. Currently there is a wealth of information presented, which might be confusing to the reader. It is suggested to concentrate on outcome measures for which there a statistically significant difference between the two groups for at least one time point. Graphical display of such measures showing their evolution with time are highly recommended. The rest of the information can be presented in a supplementary text to the main manuscript.

Author’s reply: Thank you for bringing up this point. We agree we struggle to decide how was the best way to present the data. We value a lot the opinion of the reviewers and editors, so for now we just maintained it there before hear it back from all of you. Would you suggest we transfer all tables from 2 to 5 to supplementary material? Cause we believe it is viable, although the main manuscript would lose the main results. Furthermore, we built the graphical abstract and we are showing it bellow here, to facilitate your review (it could be used also as a figure in the results):

Graphical Abstract.

Author’s reply: Well, this was really our main interest, since we hypothesized that autonomic changes and inflammatory markers could precede the blood pressure reduction that did not happen in our population. Unfortunately, in our point of view, we ending up showing that some of exercise benefits are not that easy to obtain, specially in a population of older adults that were very healthy in the baseline, with excellent medication and dietary control of their comorbidities. In any case, the readers interest of each of the variables we analyzed could have a very detailed time-course information in the tables.

Reviewer’s comment: If in the above graphical display, the statistically significant effects were added at each time point, then it would have been an excellent tool for communicating the study findings. See also previous comment, concentrating on the time points where important results occurred.

Reviewer’s comment on V2: Since the original clinical hypothesis has not been verified, the value of the work is downgraded. There might be other interesting messages in these data, but showing time-course information for so many variables might prove most of the time confusing rather than useful.

Author’s reply 2: We can delete the figure in case the reviewers do not think it is a good summary of the results anymore; although as stated previously along this review we believe it shows the only few changes that happened in a summarized way. If we remove it from the text, the readers will need to read the results of each table to find this information.

This manuscript is a resubmission of an earlier submission. The following is a list of the peer review reports and author responses from that submission.

Round 1

Reviewer 1 Report

Manuscript ID: 1706722

Title: Comprehensive time-course effects of combined training on 2 hypertensive older adults: a randomized control trial

Authors: Amanda V. Sardeli et. al.

Journal: International Journal of Environmental Research and Public Health

This paper is an interesting piece of wok. The next comments refer to the statistical analysis.

Statistical Review

  1. The authors state that two kinds of transformations were tried to achieve normality in the outcome variables: the logarithm (Log) and the square root (SqR). Both of them are special cases of a larger family of transformations, titled the Box-Cox family. For positive variables, this family is indexed by one parameter (usually called lambda in the literature), and estimation of this parameter reveals which transformation serves our purpose best. Typical examples of the Box-Cox family, apart from the Log and SqR transformations, are the Reciprocal and power transformations with integer values.
  2. Transforming to normality is not an end in itself. Today an analyst can fit a wide range of skewed or multi-modal distributions, which on the one hand they follow closely the histogram of the observed data and on the other hand they tackle appropriately the issue of outliers. Good examples are the Gamma or t distribution, or other densities within the GLM family. These models also allow the inclusion of random subject effects or other more complicated covariance structures for the repeated measurements of the response.
  3. The authors mention that both a participants random effect was included in the model, as well as a first order auto-regressive structure for the error term. This sounds like a heavy model regarding its stochastic terms. More likely to fit an independent, a compound symmetry or an AR(1) error structure and choose between them based by utilizing some well known model selection criteria (e.g. BIC). There are well established protocols for selecting terms in the systematic part of a model after deciding on the stochastic part.
  4. It seems that in the initial model, the only terms considered are group, time and the interaction group by time, alongside with the random participant effect. Baseline characteristics (see Table 1) should also be considered.
  5. P-values should be added throughout the tables
  6. Currently there is a wealth of information presented, which might be confusing to the reader. It is suggested to concentrate on outcome measures for which there a statistically significant difference between the two groups for at least one time point. Graphical display of such measures showing their evolution with time are highly recommended. The rest of the information can be presented in a supplementary text to the main manuscript.
  7. Finally since the interaction of group by time is not significant for most outcomes, then evolution of response with time might be of interest.

Reviewer 2 Report

The manuscript is on a randomized 16-week trial of combined strength and aerobic training on BP in older sedentary adults with hypertension.  Multiple mediators were examined as well.  There was no detectable difference in BP at the end of the study.  There were some changes in the mediator parameters that were examined.  There are multiple substantive concerns about the manuscript in its current form:

MAJOR

1) the sample size calculation is too vague an uninformative.  It is not clear what the magnitude of BP effect was proposed to be detected in regards to the change from baseline between the two groups.  I am not sure that the analysis used the more powerful change from baseline contrasts between the two randomized groups but rather appears to have compared the mean BP between the two groups at multiple time points - this is clearly a less powerful means of analyzing the data,

2) once the study is analyzed using the between group contrast of change from baseline, if the study results are still negative then the maximum SBP effect that could have been missed at this sample size should be explicitly stated and the 95% CIs around the between group difference in BP should be provided - to avoid throwing away all of the interim data collected prior to the end of the study, a repeated measures analysis should be employed,

3) BP was measured in the supine position - in contrast, most similar studies measure BP in the seated position.  Thus, the body position that BP was measured in is not comparable to most studies.  An aneroid device was used meaning that it was not a validated BP measurement device.  It is alos not specifically stated how many BPs were obtained and subsequently measured nor were the details of the measurement protocol described,

4) the presentation of the data is too voluminous - all of the mediators that they looked at as well as other variables beyond BP were reported at each time point across the study; the sheer number of contrasts was great and the likelihood of type I error was significant.  All of these contrasts do not need to be put in a table - focus on the BP change and put it first not buried later in the data tables.  All of these variables should be analyzed as change from baseline between the randomized treatment groups,

MINOR

1) rewrite the sentence on line 41 for improved clarity,

2) lines 47 - 48, it is not clear what "older adults have been underrepresented in hypertension guidelines"?  Under-represented in studies is understandable terminology but not guidelines,

3) lines 48 - 49, it is not clear what the statement that most adults are taking medications that limit exercise tolerance?  Older literature has reported that hypertension, per se, is associated with a 20 - 25% reduction in exercise tolerance,

4) one decimal point is enough in the text and tables when displaying BMI and percentages, and

5) statistical testing of baseline characteristics is unnecessary.